

# Prognostic role and functional impact of cadherin genes in non-small cell lung cancer tumorigenesis: mechanistic insights from *in silico* and *in vitro* analyses

Quanzhong Yang[1], Nan Feng[2], Feifei Shen[1], Lin Bai[1], Rihui Li[3], Shuang Li[1] and Weikai Zhang[1]

[1] Department of Medical Laboratory, Luoyang Polytechnic, Luoyang, China
[2] Interventional Operating Room, The First Affliated Hospital of Xinxiang Medical University, Xinxiang, China
[3] School of Basic Medicine, Sanquan College of Xinxiang Medical University, Xinxiang, China

Corresponding authors
Shuang Li, 202332005@lypt.edu.cn
Weikai Zhang, zhangweikai2021@126.com

## ABSTRACT

Non-small cell lung cancer (NSCLC) is a leading cause of cancer-related mortality worldwide, with poor prognosis and limited treatment options for advanced stages. Dysregulation of cadherin expression has been implicated in various cancers, but their exact roles and diagnostic potential of these genes in NSCLC remain unclear. The aim of this study is to investigate the diagnostic and prognostic significance of cadherin family genes (CDH1, CDH2, and CDH3) in NSCLC. This study follows an experimental design, involving both *in vitro* analyses of cell lines and survival data analysis from public databases. Nine NSCLC cell lines and five normal lung tissue-derived cell lines were cultured and CDH1, CDH2, and CDH3 expression was analyzed *via* RT-qPCR. Protein expression was validated using the Human Protein Atlas and survival analysis was conducted with the Kaplan-Meier database. Functional roles and regulatory mechanisms of cadherin genes were explored through mutational analysis, PPI networks, and miRNA interactions. The results revealed that all three cadherin genes were significantly upregulated in NSCLC cell lines and tissue samples compared to normal controls. Mutational and copy number variation analyses revealed frequent alterations in CDH2, CDH3, and CDH1 in NSCLC. Additionally, we identified hsa-miR-217, hsa-miR-203a-3p.2, and hsa-miR-6766-3p as potential regulatory miRNAs. The results of functional assays indicate that the silencing of CDH1 and CDH2 inhibits cell proliferation, colony formation, and migration in A549 cells, highlighting their potential roles in promoting tumorigenic and migratory properties in NSCLC. Collectively, our findings suggest that cadherin family genes (CDH1, CDH2, and CDH3) play critical roles in NSCLC tumorigenesis and progression, highlighting their significance as diagnostic markers.

## INTRODUCTION

Non-small cell lung cancer (NSCLC) accounts for approximately 85% of all lung cancer cases and continues to be a leading cause of cancer-related mortality worldwide (*Obeagu et al., 2023*; *Minhas, 2023*). Recent data from 2024 show that there were an estimated 2.5 million new cases of lung cancer globally, with 1.8 million deaths attributed to the disease each year (*Siegel, Giaquinto & Jemal, 2024*). NSCLC includes several subtypes, with lung adenocarcinoma (LUAD) and lung squamous cell carcinoma (LUSC) being the most prevalent, each exhibiting distinct histological and molecular features (*Liu et al., 2024*; *Hameed, 2023*). NSCLC encompasses several subtypes, including lung adenocarcinoma (LUAD) and lung squamous cell carcinoma (LUSC), each with distinct histological and molecular features (*Shen, Chen & Li, 2024*; *Li, Ma & Al-Obeidi, 2024*). Despite advancements in diagnostic technologies, such as low-dose computed tomography (LDCT) for early detection and molecular profiling for targeted therapies, the overall survival rate for NSCLC patients remains unacceptably low. For patients diagnosed at advanced stages, the 5-year survival rate is less than 20%, largely due to late-stage diagnosis and the aggressive nature of the disease (*Song, Kelava & Kiss, 2023*). Therapeutic advancements, including tyrosine kinase inhibitors (TKIs) targeting EGFR mutations or ALK rearrangements and immune checkpoint inhibitors targeting PD-1/PD-L1, have revolutionized NSCLC treatment (*Parvaresh et al., 2024*; *Liao et al., 2022*). However, these therapies are effective only in a limited subset of patients, as many individuals lack actionable genetic alterations or develop resistance to these treatments over time (*Liao et al., 2022*). Moreover, the heterogeneous nature of NSCLC, both intertumoral and intratumoral, poses significant challenges to treatment efficacy (*Ottaiano et al., 2023*; *Munteanu et al., 2023*; *Wang et al., 2024*). Therefore, identifying novel diagnostic and prognostic targets is essential to address these gaps, providing opportunities for early detection, patient stratification, and the development of more effective treatment strategies tailored to individual tumor profiles.

The Cadherin family of genes, including CDH1 (E-cadherin), CDH2 (N-cadherin), and CDH3 (P-cadherin), plays a vital role in regulating cell-cell adhesion, maintaining tissue architecture, and controlling cellular processes such as differentiation, migration, and proliferation (*Yu et al., 2019*; *Garg & Singh, 2019*; *Lungu et al., 2024*). Cadherins are transmembrane glycoproteins that mediate calcium-dependent adhesion between cells, providing structural integrity to tissues (*Hirano & Takeichi, 2012*). CDH1 is primarily expressed in epithelial cells and plays a critical role in maintaining epithelial integrity and preventing epithelial-mesenchymal transition (EMT) (*Noronha et al., 2024*). CDH2 is predominantly found in neural and muscle tissues, while CDH3 is expressed in placental and epithelial cells (*Gheldof & Berx, 2013*). Dysregulation of cadherins, particularly E-cadherin, is implicated in the loss of cell adhesion, leading to increased cellular motility, invasion, and metastasis, which are hallmarks of cancer progression (*Bryan, 2015*; *Sousa, Pereira & Paredes, 2019*; *Lin, Cooper & Anastasiadis, 2023*). Moreover, cadherin genes have been shown to exhibit altered expression in NSCLC, affecting cell-cell adhesion and contributing to the epithelial-to-mesenchymal transition (EMT), a key process in cancer

metastasis (*Tan et al., 2021*; *Ito et al., 2025*). In NSCLC, the downregulation or loss of CDH1 (*Fan et al., 2019*; *Lofiego et al., 2021*) and upregulation of CDH2 is frequently observed and is associated with increased invasiveness and the ability of tumor cells to detach from the primary tumor site. However, the role of CDH3 is not widely studied in NSCLC.

This study aims to investigate the diagnostic and prognostic significance of the cadherin family genes, including CDH1, CDH2, and CDH3, in NSCLC. Using an integrated approach involving *in silico* analyses (*Jiang et al., 2025*) and *in vitro* experimental validation (*Hameed & Ejaz, 2021*), we will explore the expression patterns, molecular interactions, and clinical relevance of these genes in NSCLC. Functional assays will be employed to assess the impact of cadherin modulation on cell proliferation, migration, and survival. Given the established role of cadherins in other cancers, this study seeks to provide novel insights into the mechanistic role of cadherin family genes in NSCLC, offering potential avenues for targeted interventions.

## METHODOLOGY

### Cell culture

Nine NSCLC cell lines (A549, H1299, H1975, H460, H292, H358, H661, H520, and H226) and five normal fibroblast cell lines (BEAS-2B, NL-20, MRC-5, WI-38, and CCD-19Lu) were purchased from ATCC, USA. The NSCLC cell lines represent a range of molecular subtypes, including A549 (KRAS-mutant adenocarcinoma), H1299 (p53-null adenocarcinoma), H1975 (EGFR L858R and T790M mutations), H460 (large cell carcinoma), H292 (mucinous adenocarcinoma), H358 (EGFR-positive adenocarcinoma), H661 (adenocarcinoma), H520 (squamous cell carcinoma), and H226 (squamous cell carcinoma). NSCLC cell lines were cultured in RPMI-1640 medium supplemented with 10% fetal bovine serum (FBS) and 1% penicillin-streptomycin. Normal lung cell lines were grown under specific conditions: BEAS-2B and NL-20 in bronchial epithelial cell basal medium (BEGM) with growth supplements, and MRC-5, WI-38, and CCD-19Lu in DMEM with 10% FBS and 1% penicillin-streptomycin. All cell lines were maintained at 37 °C in a humidified incubator with 5% $CO_2$. The study was conducted at Luoyang Polytechnic, Luoyang 471000, China.

### Reverse transcription quantitative PCR (RT-qPCR)

Total RNA was extracted from all cell lines using the TRIzol$^{TM}$ reagent (Invitrogen, Waltham, MA, USA), following the manufacturer's protocol. Briefly, cells were harvested and lysed in TRIzol reagent, which facilitates the simultaneous extraction of RNA, DNA, and proteins. The RNA was then separated from the other cellular components by adding chloroform, followed by centrifugation. The aqueous phase containing the RNA was carefully collected, and isopropanol was added to precipitate the RNA. The RNA pellet was washed with 75% ethanol to remove any impurities and then resuspended in RNAse-free water. RNA purity and concentration were assessed using a NanoDrop$^{TM}$ spectrophotometer (Thermo Fisher Scientific, Waltham, MA, USA), and samples with an

A260/A280 ratio of 1.8–2.0 were selected for further analysis. Complementary DNA (cDNA) was synthesized from 1 μg of RNA using the High-Capacity cDNA Reverse Transcription Kit (Applied Biosystems, Waltham, MA, USA), according to the kit instructions. RT-qPCR of CDH1, CDH2, and CDH3 was then performed using PowerUp™ SYBR™ Green Master Mix (Applied Biosystems, Waltham, MA, USA) on a QuantStudio™ 5 Real-Time PCR System (Thermo Fisher Scientific, Waltham, MA, USA). Relative gene expression was calculated using the $2^{-\Delta\Delta Ct}$ method. Following primers were used for the amplification purpose.

GAPDH-F 5′-ACCCACTCCTCCACCTTTGAC-3′,
GAPDH-R 5′-CTGTTGCTGTAGCCAAATTCG-3′
CDH1-F: 5′-GCCTCCTGAAAAGAGAGTGGAAG-3′
CDH1-R: 5′-TGGCAGTGTCTCTCCAAATCCG-3′
CDH2-F: 5′-CCTCCAGAGTTTACTGCCATGAC-3′
CDH2-R: 5′-GTAGGATCTCCGCCACTGATTC-3′
CDH3-F: 5′-CAGGTGCTGAACATCACGGACA-3′
CDH3-R: 5′-CTTCAGGGACAAGACCACTGTG-3′

## Validation of expression, enrichment, and functional roles of cadherin family genes in extended NSCLC cohorts

The GSCA database (*Liu et al., 2023*) was employed in this study to investigate the expression, enrichment, and functional roles of cadherin genes in NSCLC. This database integrates multi-omics datasets, including The Cancer Genome Atlas (TCGA) and Genotype-Tissue Expression (GTEx), enabling systematic exploration of gene roles in various cancer types. Differential expression analysis was performed using extended NSCLC TCGA cohorts to compare tumor and normal tissues. Gene Set Enrichment Analysis (GSEA) was conducted *via* the GSCA database to identify the involvement of cadherin genes in LUAD and LUSC pathogenesis. Furthermore, gene expression was analyzed across different pathological stages to evaluate their correlation with tumor progression. Functional pathway analysis was carried out to explore the activation of key cancer-related pathways due to the dysregulation of cadherin genes.

## Protein expression validation and prognostic implications of cadherin genes in NSCLC

Protein expression levels of cadherin family genes were analyzed using the Human Protein Atlas (HPA) database (*Thul & Lindskog, 2018*). Immunohistochemical (IHC) staining data were retrieved to compare protein levels in normal lung tissues and NSCLC tissues. For survival analysis, the Kaplan-Meier (KM) Plotter database was employed to evaluate the prognostic significance of CDH1, CDH2, and CDH3 (*Lánczky & Győrffy, 2021*). Hazard ratios (HRs) and *p*-values were calculated to determine the association between gene expression and overall survival.

## Mutational landscape, copy number variation (CNV), and promoter methylation status of cadherin genes in NSCLC

The mutational analysis of cadherin family genes (CDH1, CDH2, and CDH3) in NSCLC was conducted using the cBioPortal database (*Cerami et al., 2012*). This included the evaluation of mutation frequency, variant types, and specific mutation patterns. Furthermore, CNV data were also retrieved from cBioPortal, assessing the prevalence of heterozygous amplifications and deletions in NSCLC samples. Promoter methylation status of cadherin genes was analyzed using the OncoDB database (*Tang, Cho & Wang, 2022*) to compare methylation levels between NSCLC tumor tissues and normal tissues.

## Correlation of cadherin gene expression with molecular subtypes and immune inhibitory molecules in NSCLC

The expression levels of cadherin genes across six molecular subtypes (C1 to C6) of NSCLC were analyzed using the TISIDB database (*Ru et al., 2019*). Subtype-specific variations in gene expression were evaluated to explore the role of cadherin genes in tumor heterogeneity and clinical behavior. Additionally, correlations between cadherin gene expression and immune inhibitory molecules were assessed across various cancers, with a focus on LUAD and LUSC.

## Regulatory role and diagnostic potential of miRNAs targeting cadherin genes in NSCLC

To explore the regulatory role of microRNAs (miRNAs) targeting cadherin genes in NSCLC, we utilized the HumanTargetScan database (*Agarwal et al., 2015*). TargetScan is a widely used tool for predicting miRNA targets by analyzing the sequence complementarity between miRNAs and the 3′ untranslated regions (3′ UTRs) of target genes. The database provides a comprehensive list of miRNAs that potentially regulate gene expression at the post-transcriptional level by binding to specific regions within the target gene's 3′ UTR, leading to mRNA degradation or translational repression. This analysis identified hsa-miR-217 as a predicted regulator of CDH1, hsa-miR-203a-3p.2 as a regulator of CDH2, and hsa-miR-6766-3p as a regulator of CDH3.

The expression of these miRNAs was experimentally validated by performing RT-qPCR analysis on RNA extracted from 9 NSCLC cell lines and five normal control cell lines. Total RNA, including miRNA, was extracted from NSCLC cell lines and normal control cell lines using the miRNeasy Mini Kit (Qiagen, Germantown, MD, USA) following the manufacturer's instructions. The extracted RNA was reverse-transcribed into cDNA using the TaqMan Advanced miRNA cDNA Synthesis Kit (Applied Biosystems, Waltham, MA, USA). RT-qPCR was then performed using TaqMan Advanced miRNA Assays (Applied Biosystems, Waltham, MA, USA) specific for hsa-miR-217, hsa-miR-203a-3p.2, hsa-miR-6766-3p, and U6 (internal control) on a QuantStudio 5 Real-Time PCR System (Applied Biosystems, USA). Relative expression levels were calculated using the $2^{-\Delta\Delta Ct}$ method, normalizing to U6 small nuclear RNA as the endogenous control.

## PPI networks and functional enrichment analysis of cadherin family genes

Protein-protein interaction (PPI) networks for cadherin genes were constructed using the GeneMANIA (*Warde-Farley et al., 2010*) and STRING (*Szklarczyk et al., 2023*) databases to reveal their interconnected relationships with other proteins. Furthermore, the common interacting genes between PPIS were identified through a Venn diagram analysis. Moreover, to explore the biological significance of cadherin genes interacting partners in NSCLC, gene enrichment analysis was performed using the DAVID tool (*Jia, Xu & Wang, 2021*), focusing on cellular components, molecular functions, biological processes, and pathways enrichment.

DAVID is a comprehensive bioinformatics tool that facilitates functional annotation and enrichment analysis of large gene sets.

## Association of cadherin family gene expression with immune cell infiltration and drug sensitivity in NSCLC

To explore the relationship between cadherin family gene expression and immune cell infiltration as well as drug sensitivity in NSCLC, we utilized the GSCA database (*Liu et al., 2023*). The GSCA database is an integrative platform that provides comprehensive analyses of gene expression, mutation profiles, immune infiltration, and drug sensitivity across a wide range of cancers.

## CDH1 and CDH2 gene knockdown in A549 cells

For the knockdown of CDH1 and CDH2, siRNAs targeting these genes were purchased from Thermo Fisher. The siRNA sequences used were: CDH1 siRNA (5′-CAGACAAAGA CCAGGACTA-3′) and CDH2 siRNA (5′-GGACCCAGAUCGAUAUAUGTT-3′). A549 cells were cultured in RPMI-1640 medium supplemented with 10% FBS and incubated at 37 °C in a 5% $CO_2$ atmosphere. To perform the transfection, 10 nM of the respective siRNA was mixed with Lipofectamine™ RNAiMAX Transfection Reagent (Thermo Fisher, Waltham, MA, USA) in Opti-MEM medium according to the manufacturer's instructions. After a 5-min incubation at room temperature, the transfection mixture was added to A549 cells that were grown to 70–80% confluence. Cells were incubated for 48 h, after which they were harvested for RNA extraction and analyzed by RT-qPCR to confirm the knockdown efficiency of CDH1 and CDH2.

For confirming successful knockdown of CDH1 and CDH2, we further performed RT-qPCR and Western blot analyses. RT-qPCR of CDH1 and CDH2 was performed using after mentioned conditions, while for Western blot, protein lysates were collected from the transfected A549 cells using RIPA buffer (Thermo Fisher, Waltham, MA, USA) containing protease and phosphatase inhibitors. Protein concentrations were measured using the BCA Protein Assay Kit (Thermo Fisher, Waltham, MA, USA). Equal amounts of protein (30 μg) were separated by SDS-PAGE (10% gel) and transferred onto a PVDF membrane (Millipore, Burlington, MA, USA). The membrane was blocked with 5% BSA (Sigma-Aldrich, St. Louis, MO, USA) in TBS-T for 1 h at room temperature. The membrane was then incubated overnight at 4 °C with primary antibodies against CDH1

(1:1000; Abcam, Cambridge, UK), CDH2 (1:1000; Abcam, Cambridge, UK), and GAPDH (1:5000; Abcam, Cambridge, UK) as a loading control. After washing, the membrane was incubated with a HRP-conjugated secondary antibody (1:5000, Thermo Fisher, Waltham, MA, USA) for 1 h at room temperature. Detection was performed using the ECL Western Blotting Substrate (Thermo Fisher, Waltham, MA, USA), and protein bands were visualized and quantified using the ChemiDoc Imaging System (Bio-Rad, Hercules, CA, USA). The intensity of protein bands was normalized to GAPDH. The densitometric analysis of the protein bands was performed using Image Lab Software (Bio-Rad, Hercules, CA, USA). The band densities were calculated by measuring the pixel intensity within the defined area of each protein band. Background subtraction was applied by measuring the intensity in a region adjacent to the band of interest. The relative band intensity for each target protein was normalized to the corresponding GAPDH band intensity to account for any variations in protein loading.

## Cell proliferation assay

To evaluate the effect of CDH1 and CDH2 knockdown on cell proliferation, we performed the Cell Counting Kit-8 (CCK-8) assay (Dojindo, Mashiki, Kumamoto, Japan). A549 cells were seeded into 96-well plates ($1 \times 10^3$ cells/well) and allowed to adhere overnight. After 24, 48, and 72 h of siRNA transfection, 10 μL of CCK-8 solution was added to each well, and the cells were incubated at 37 °C for 2 h. Absorbance was measured at 450 nm using a Microplate Reader (Thermo Fisher, Waltham, MA, USA). The relative cell proliferation rate was calculated by normalizing the absorbance values to the control group (non-transfected cells).

## Colony formation assay

To assess long-term cell proliferation (Qiu et al., 2024), we performed a colony formation assay. After 48 h of siRNA transfection, A549 cells were trypsinized and seeded at a density of 500 cells per 6-well plate. The cells were incubated for 10–14 days, with the medium replaced every 3 days. After incubation, cells were fixed with 4% paraformaldehyde for 30 min at room temperature, followed by staining with 0.5% crystal violet solution (Sigma-Aldrich, St. Louis, MO, USA) for 15 min. Colonies (defined as ≥40 cells) were counted under a light microscope, and the number of colonies in each group was recorded.

## Wound healing assay

The wound healing assay (Yang et al., 2025) was performed to assess the migratory capacity of A549 cells following CDH1 and CDH2 knockdown. After 48 h of siRNA transfection, A549 cells were seeded in a 6-well plate ($1 \times 10^5$ cells/well) and allowed to form a monolayer. A sterile P200 pipette tip was used to create a uniform wound across the cell monolayer. The cells were washed with phosphate-buffered saline (PBS) to remove debris, and fresh medium was added. Images of the wound area were captured at 0 and 24 h using a light microscope (Olympus, Shinjuku City, Tokyo, Japan) at 10× magnification. The wound area was measured using ImageJ software (NIH, Bethesda, MD,

USA), and the wound healing percentage was calculated by comparing the reduction in wound area at different time points relative to the initial wound area.

## Statistics

Statistical analyses were conducted using GraphPad Prism 9.0 software. Differences between experimental groups were analyzed using a one-way analysis of variance (ANOVA) followed by Tukey's *post hoc* test for multiple comparisons. For RT-qPCR results, gene expression levels were calculated using the $2^{-\Delta\Delta Ct}$ method, and statistical significance was determined by unpaired t-tests when comparing two groups. Kaplan-Meier survival analysis was used to assess the prognostic significance of KCTD genes in NSCLC, with hazard ratios and log-rank tests calculated to determine significance. $P$*-value < 0.05, $P$**-value < 0.01, and $P$***-value < 0.001 were considered statistically significant.

# RESULTS

## Expression and diagnostic potential of cadherin genes in NSCLC and normal control cell lines

In the first part of the study, we investigated the expression levels of CDH1, CDH2, and CDH3 across nine NSCLC cell lines and five normal control cell lines using RT-qPCR technique. The analysis revealed that all three cadherin genes were significantly upregulated in NSCLC cell lines compared to the normal control cell lines (Fig. 1A). To evaluate the diagnostic potential of these genes using RT-qPCR expression data, we conducted ROC curve analysis. The input data consisted of normalized expression values obtained from NSCLC and matched normal cell line samples. The ROC curves were generated using the pROC package in R, where sensitivity and specificity were calculated to assess the discriminatory power of each gene. Notably, the area under the curve (AUC) for CDH1, CDH2, and CDH3 was 1.0 in all three cases, indicating perfect sensitivity and specificity in this dataset (Fig. 1B). The plots appear as straight lines due to the binary nature of the dataset and the complete separation of sample groups based on gene expression, reflecting excellent diagnostic performance (Fig. 1B).

## Expression validation across extend cohorts, enrichment, and functional roles of cadherin family genes in NSCLC

Next, we investigated the expression, enrichment, and functional roles of cadherin genes (CDH1, CDH2, and CDH3) in LUAD and LUSC using data from the GSCA database. Expression validation using extended NSCLC TCGA cohorts showed that CDH1, CDH2, and CDH3 genes were significantly upregulated in both LUAD and LUSC tumors compared to normal tissues (Figs. 2A, 2B). Next, the GSEA analysis was conducted using a gene set comprising CDH1, CDH2, and CDH3 to investigate their collective role in lung cancer. The analysis was performed on LUAD and LUSC datasets, and key parameters including the normalized enrichment score (NES), adjusted $P$-value, and false discovery rate (FDR) were calculated to ensure statistical validity. In LUAD, the cadherin gene set showed a strong enrichment with NES score of 1.85, a nominal $P$-value of < 0.001, and an

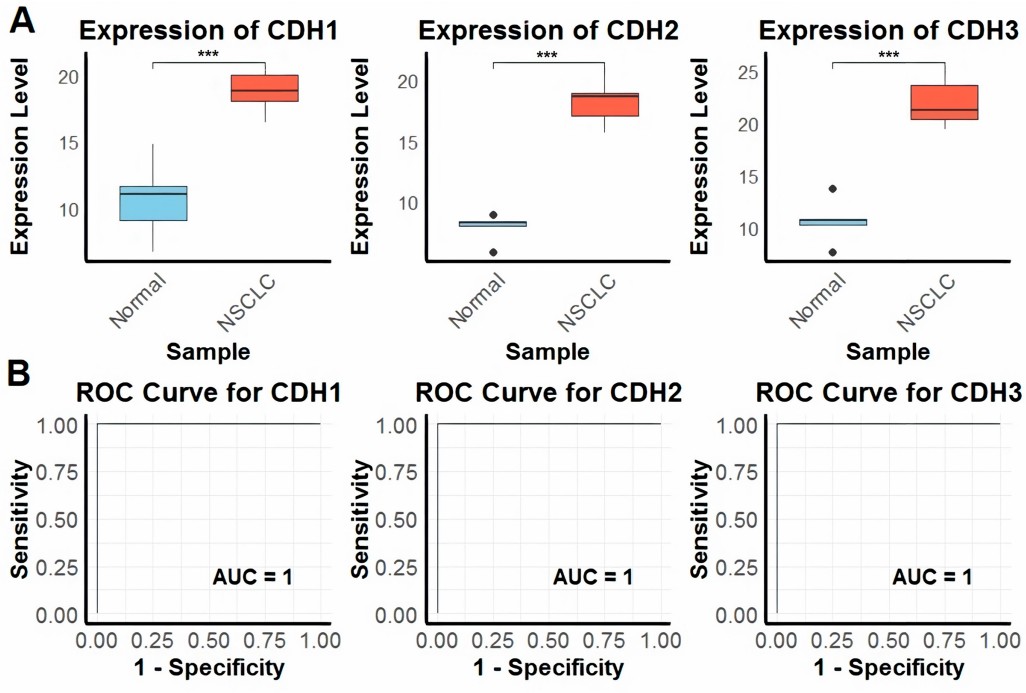

**Figure 1 Expression and diagnostic accuracy of CDH1, CDH2, and CDH3 in NSCLC cell lines.** (A) RT-qPCR analysis of CDH1, CDH2, and CDH3 expression levels in NSCLC cell lines ($n = 9$) and normal control cell lines ($n = 5$). (B) Receiver operating characteristic (ROC) curve analysis demonstrating the diagnostic potential of CDH1, CDH2, and CDH3, with area under the curve (AUC) values of 1.0 for each gene. $P$***-value $< 0.001$.

FDR of 0.01. Similarly, in LUSC, the NES was 1.80, with a nominal $P$-value of $< 0.001$ and an FDR of 0.02. These results demonstrate significant enrichment of the cadherin gene set in tumor samples compared to normal tissues, highlighting their potential involvement in the pathogenesis of both LUAD and LUSC (Figs. 2C, 2D). Moreover, analysis of gene expression across pathological stages showed consistent upregulation of CDH1, CDH2, and CDH3 in advanced tumor stages, emphasizing the potential role of cadherin genes in NSCLC progression (Figs. 2E, 2F). Furthermore, the functional pathway analysis of cadherin genes reveled their involvement in the activation of various important pathways. For instance, CDH2 and CDH3 were found activating epithelial-to-mesenchymal transition (EMT) and Cell cycle cancer-related pathways in NSCLC (Fig. 2F).

## Protein expression validation and prognostic significance of cadherin family genes in NSCLC

In this part of our study, we validate the protein expression levels and prognostic significance of cadherin family genes in NSCLC using HPA and KM plotter databases. The immunohistochemical analysis revealed that in normal tissues, CDH1, CDH2, and CDH3 exhibited low staining intensity, indicating low protein levels (Fig. 3A). In contrast, LUAD tissues showed high staining intensity for all three genes, confirming their elevated protein expression in tumors (Fig. 3A). Survival analysis using the KM Plotter tool, with patients

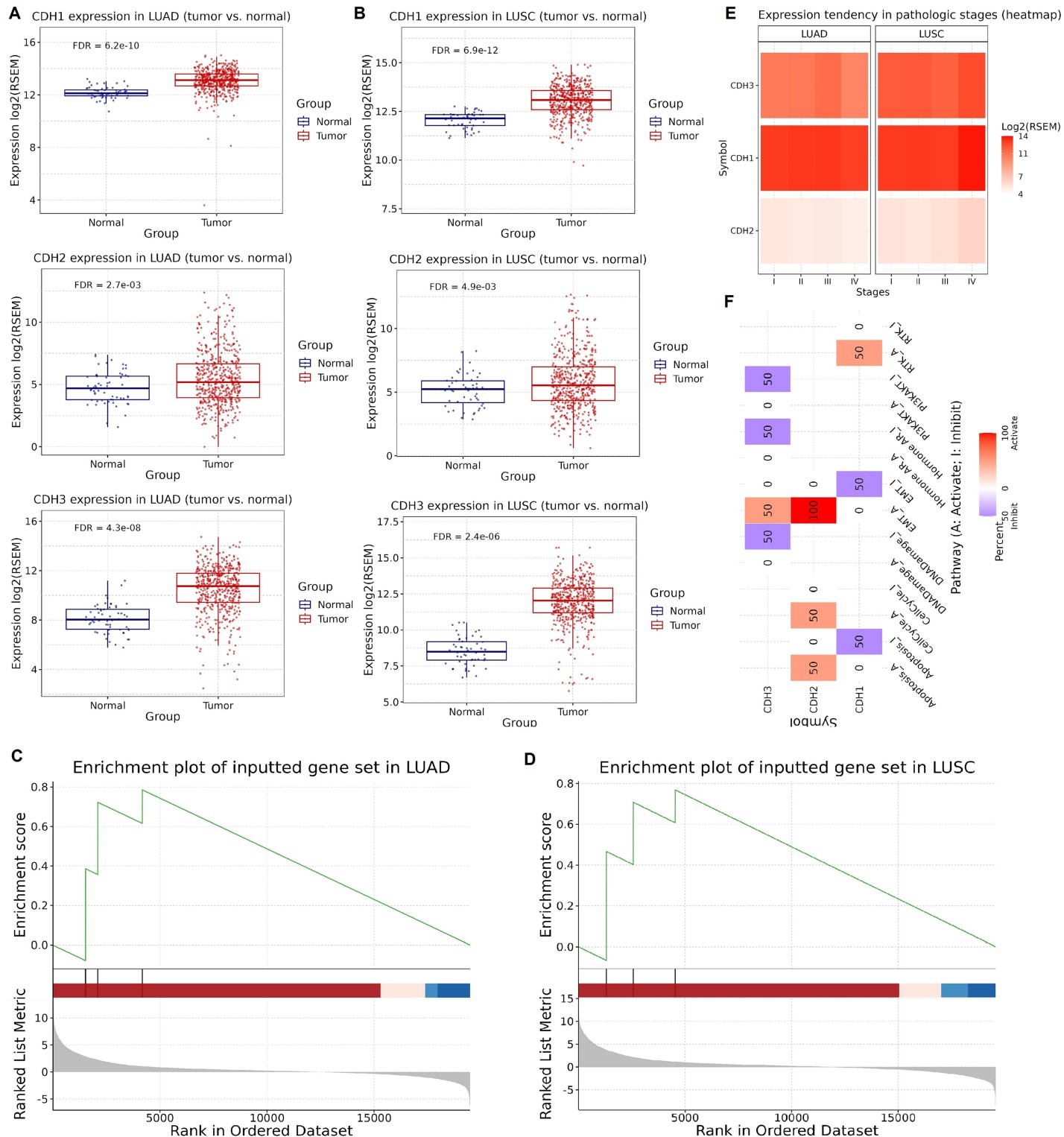

**Figure 2 Expression validation, GSEA, and pathway analysis of cadherin genes in LUAD and LUSC.** (A and B) Expression validation analysis of CDH1, CDH2, and CDH3 using extended TCGA NSCLC cohorts. (C and D) GSEA highlighted the involvement of cadherin family genes in the pathogenesis of LUAD and LUSC. (E) Expression levels of CDH1, CDH2, and CDH3 across pathological stages of NSCLC. (F) Functional pathway analysis of cadherin genes in NSCLC. *P*-value < 0.05.

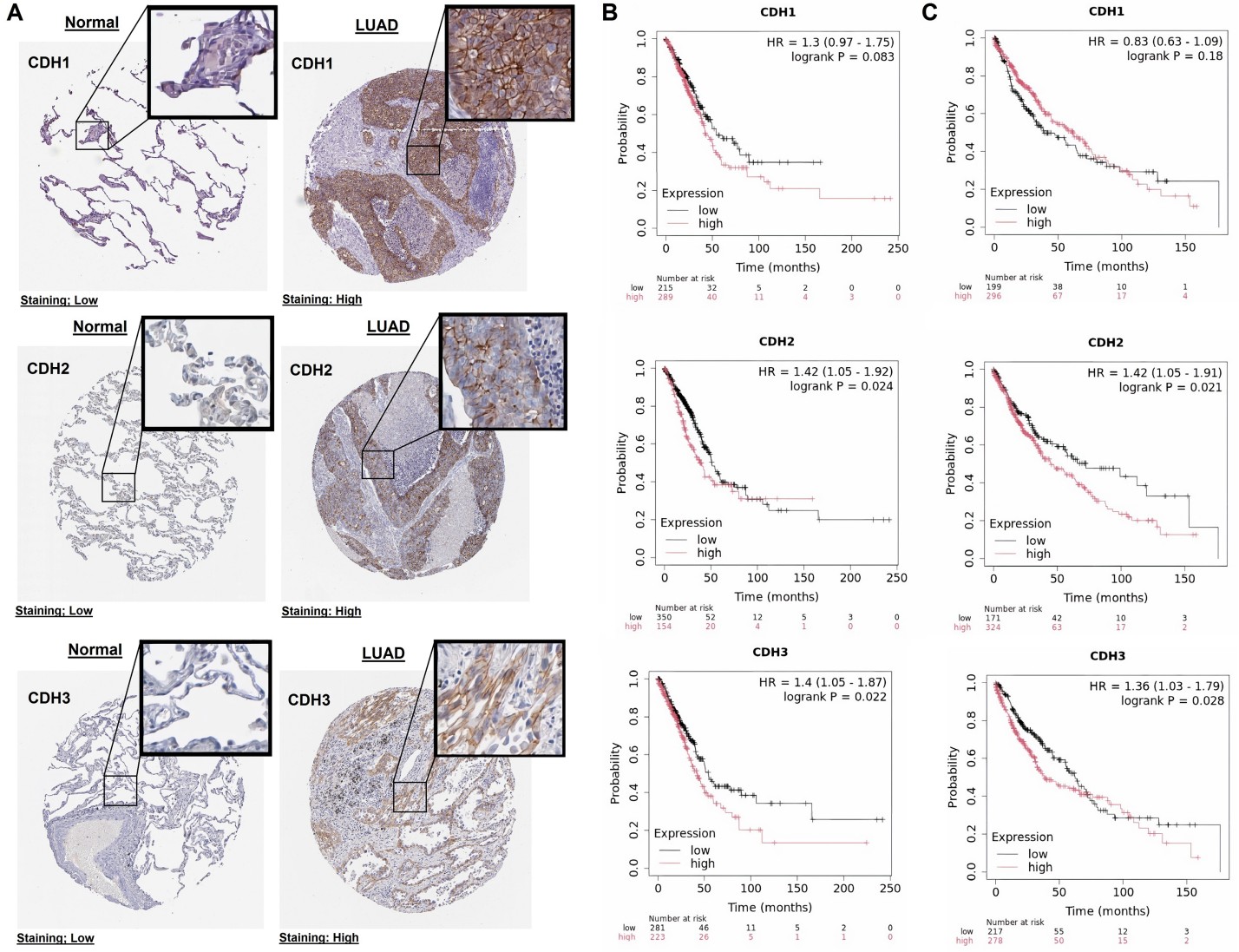

**Figure 3 Protein expression and prognostic significance of cadherin genes in NSCLC.** (A) Immunohistochemical analysis of CDH1, CDH2, and CDH3 protein levels using the HPA database. (B) Overall survival analysis of cadherin genes in LUAD using the KM plotter database. (C) Overall survival analysis of cadherin genes in LUSC using the KM plotter database. *P*-value < 0.05.

selected based on auto best cutoff, overall survival as the endpoint, and split by median expression, indicated a trend toward poorer overall survival with high CDH1 expression in LUAD (HR = 1.3, *P* = 0.083), though it was not statistically significant (Fig. 3B). Similarly, in LUSC, no significant association was observed between CDH1 expression and overall survival of LUSC patients (Fig. 3C). The lack of significance may be due to limited sample size, and larger cohorts or meta-analyses may be needed to validate this potential prognostic role. High expression of CDH2 and CDH3 was significantly associated with poorer overall survival in NSCLC patients (Figs. 3B, 3C). CDH2 showed hazard ratios of 1.42 (*P* = 0.024) in LUAD and 1.42 (*P* = 0.021) in LUSC, while CDH3 had hazard ratios of 1.40 (*P* = 0.022) in LUAD and 1.36 (*P* = 0.028) in LUSC. These findings suggest that

elevated expression of these genes correlates with more aggressive tumor behavior and worse prognosis.

## Mutational landscape, CNV, and promoter methylation analysis of cadherin family genes in NSCLC

In this section, we investigated the mutational landscape, CNV, and promoter methylation status of cadherin family genes in NSCLC. The mutational analysis *via* cBioPorta database revealed that cadherin genes were frequently altered in NSCLC samples. Among the analyzed samples, CDH2 was the most frequently mutated gene, with mutations identified in 63% of cases, followed by CDH3 and CDH1, which were altered in 23% and 19% of cases, respectively (Fig. 4A). The majority of these alterations were missense mutations, with additional occurrences of nonsense mutations, splice site mutations, and frameshift deletions (Fig. 4B). Single nucleotide polymorphism (SNP) was the predominant variant type. Transition mutations (Ti) were more common than transversions (Tv), with C > T and T > G transitions being the most frequently observed (Fig. 4C).

The CNV analysis further showed that heterozygous amplification and deletion were the common types of CNVs in NSCLC patients (Fig. 4D). Furthermore, promoter methylation analysis of cadherin genes was performed *via* the OncoDB to evaluate the epigenetic regulation of CDH1, CDH2, and CDH3 in NSCLC. The results indicated differential methylation patterns between tumor and normal tissues. In LUAD, CDH1 and CDH3 exhibited hypermethylation at the promoter regions, while CDH2 showed moderate methylation changes (Fig. 4E). Similar trends were observed in LUSC, with hypermethylation of CDH1 and CDH3 promoters and relatively stable methylation levels for CDH2 (Fig. 4E).

## Correlation of cadherin gene expression with molecular subtypes and immune inhibitory molecules in NLSC

In the next part, we analyzed the expression of cadherin genes across different molecular subtypes of NSCLC, including LUAD and LUSC, using the TISIDB database. Additionally, we investigated the correlation between cadherin gene expression and immune inhibitory molecules in various cancers, with a focus on LUAD and LUSC, to explore their potential role in tumor immunity and progression. The expression analysis revealed significant differences in cadherin gene (CDH1, CDH2, and CDH3) levels among the 5 molecular subtypes (C1, C2, C3, C4, and C6) in both LUAD and LUSC (Figs. 5A, 5B). Notably, subtype C5 was not included in this analysis as it was absent in the dataset used, which may be due to insufficient sample size or lack of classification data for this subtype within the selected cohorts. CDH1 expression was higher in C1, suggesting a role in maintaining cell-cell adhesion and epithelial-like characteristics, indicating better tumor differentiation (Figs. 5A, 5B). In contrast, CDH2 was upregulated in C6, a mesenchymal-like subtype associated with increased invasion and metastasis, suggesting greater aggressiveness (Figs. 5A, 5B). CDH3 expression was higher in C4 in LUAD, which may contribute to tumor progression and therapy resistance, while in C2 in LUSC, it could support tumor proliferation and resistance mechanisms. These findings show that cadherins are expressed

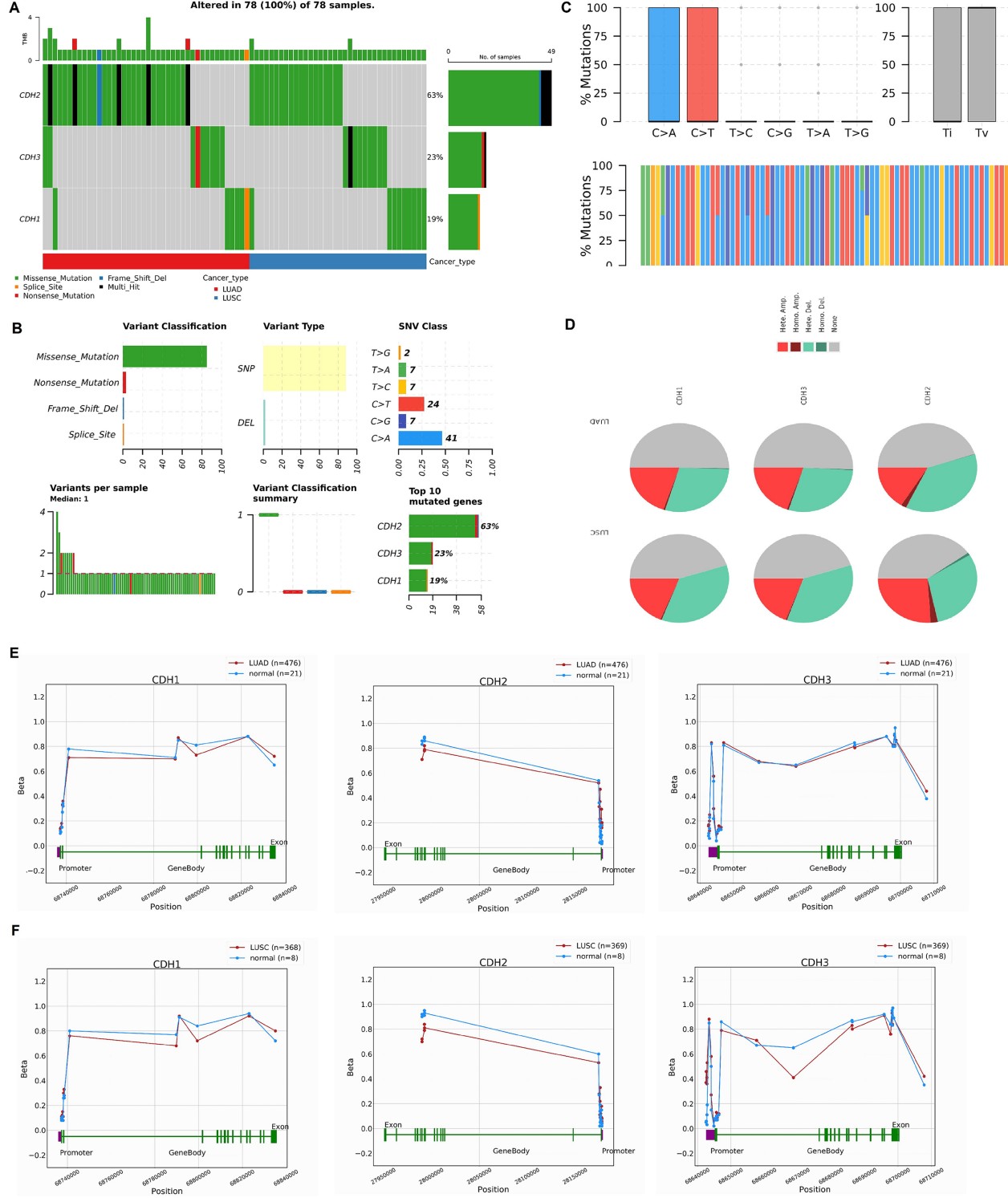

**Figure 4  Mutational landscape, copy number variations (CNVs), and promoter methylation analysis of cadherin genes in NSCLC.** (A) Mutational analysis of cadherin genes (CDH1, CDH2, and CDH3) in NSCLC samples using the cBioPortal database. (B) Types of detected mutations in cadherin genes across NSCLC samples. (C) Types of detected SNP in cadherin genes across NSCLC samples. (D) CNVs analysis of cadherin genes across NSCLC samples. (E) Promoter methylation analysis of CDH1, CDH2, and CDH3 in LUAD tumors using the OncoDB database. (F) Promoter methylation analysis of CDH1, CDH2, and CDH3 in LUSC tumors using the OncoDB database. *P*-value < 0.05.

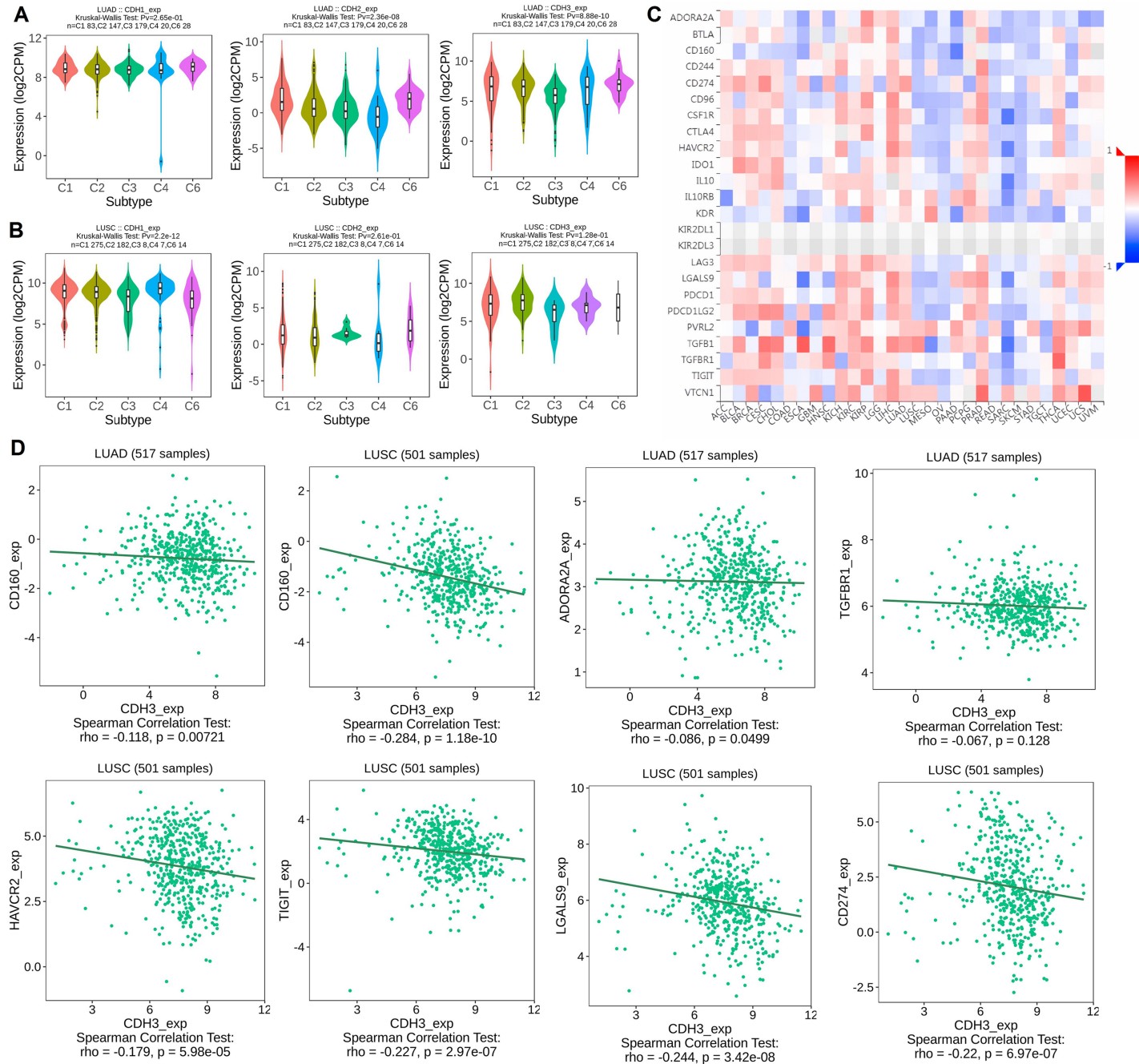

**Figure 5 Expression of cadherin genes across NSCLC molecular subtypes and their correlation with immune inhibitory molecules.** (A, B) Expression analysis of cadherin genes (CDH1, CDH2, and CDH3) across six molecular subtypes (C1 to C6) of NSCLC. (C) Heatmap analysis illustrating the correlation between cadherin gene expression and immune inhibitory molecules across various cancers. (D) Correlation between cadherin gene expression and selected immune inhibitory molecules. *P*-value < 0.05.

differently across NSCLC subtypes, with subtype-specific roles that could impact tumor progression, metastasis, and therapeutic resistance. This emphasizes the need for personalized treatment approaches targeting the unique molecular features of each subtype.

To understand the immune implications of cadherin expression, we examined its correlation with immune inhibitory genes across various cancers, as visualized in the heatmap analysis (Fig. 5C). In LUAD and LUSC, differential correlations were observed between cadherin genes and key immune inhibitors. For example, In LUAD, CDH3 expression was negatively correlated with immune inhibitory genes such as CD160 (Spearman's rho = −0.118, $p$ = 0.00721), ADORA2A (rho = −0.086, $p$ = 0.0499), and TGFB1 (rho = −0.067, $p$ = 0.128, non-significant). In LUSC, significant negative correlations were observed between CDH3 and CD160 (rho = −0.284, $p = 1.18 \times 10^{-10}$), HAVCR2 (rho = −0.227, $p = 2.97 \times 10^{-7}$), TIGIT (rho = −0.244, $p = 3.42 \times 10^{-8}$), LGALS9 (rho = −0.22, $p = 6.97 \times 10^{-7}$), and CD274/PD-L1 (rho = −0.179, $p = 5.98 \times 10^{-5}$) (Fig. 5D).

## Regulatory role and diagnostic potential of miRNAs targeting cadherin genes in NSCLC

Next, we evaluated regulatory microRNAs (miRNAs) targeting cadherin family genes (CDH1, CDH2, and CDH3) in NSCLC using the HumanTargetScan database. The analysis identified hsa-miR-217 as a predicted regulator of CDH1, hsa-miR-203a-3p.2 as a predicted regulator of CDH2, and hsa-miR-6766-3p as a predicted regulator of CDH3 (Fig. 6A). These predictions were based on conserved binding sites, context++ scores, and predicted binding affinity. For example, hsa-miR-217 was predicted to bind to the CDH1 3′ UTR at position 125–131 with a context++ score of −0.16 and a weighted context++ score of −0.16, indicating a high-confidence interaction (Fig. 6A).

To validate the relevance of these miRNAs in NSCLC, we performed RT-qPCR analysis to compare their expression levels in nine NSCLC cell lines and five normal control cell lines. The results showed significant downregulation of hsa-miR-217, hsa-miR-203a-3p.2, and hsa-miR-6766-3p in NSCLC cell lines compared to normal controls ($p$ < 0.01 for all comparisons) (Fig. 6B). We further evaluated the diagnostic potential of these miRNAs using ROC curve analysis. The AUC values for hsa-miR-217, hsa-miR-203a-3p.2, and hsa-miR-6766-3p were 0.902, 0.823, and 0.927, respectively (Fig. 6C). These results indicate excellent diagnostic performance for hsa-miR-217 and hsa-miR-6766-3p, and good diagnostic performance for hsa-miR-203a-3p.2, in distinguishing NSCLC from normal controls.

## PPI networks and functional enrichment analysis of cadherin family genes

In this section of the study, the interaction networks and functional roles of cadherin family genes in NSCLC were analyzed. Firstly, the PPI networks of cadherin genes (CDH1, CDH2, and CDH3) were constructed using the GeneMANIA and STRING databases (Figs. 7A, 7B), and common interacting genes were identified through a Venn diagram analysis (Fig. 7C). The PPI networks and shared interacting partners revealed that cadherin proteins interact with other cadherins and members of the catenin family, which are key regulators of cell adhesion and intracellular signaling. This finding is consistent with previous studies highlighting the importance of cadherin-cadherin and cadherin-catenin interactions in cancer progression, particularly in the regulation of EMT

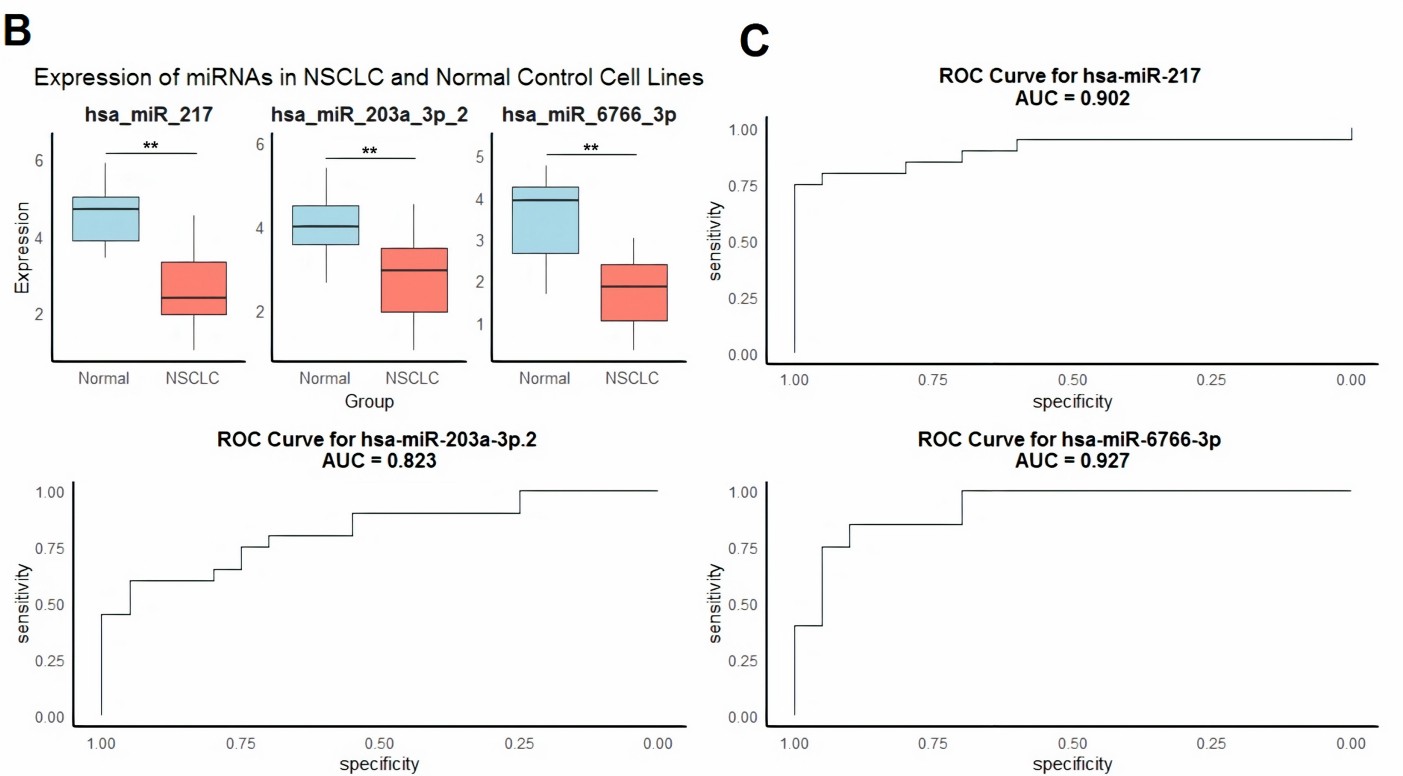

**Figure 6 Regulatory microRNAs targeting cadherin genes and their expression in NSCLC.** (A) Predicted regulatory microRNAs for cadherin family genes identified using the HumanTargetScan database. (B) RT-qPCR analysis of hsa-miR-217, hsa-miR-203a-3p.2, and hsa-miR-6766-3p expression in 9 NSCLC cell lines and five normal control cell lines. (C) ROC analysis of the predicted miRNAs. $P$**-value < 0.01.

and metastasis (*Conacci-Sorrell et al., 2003*; *Lin, Cooper & Anastasiadis, 2023*). Importantly, the identification of common genes between the two networks was critical to understanding which signaling pathways may be disrupted due to cadherin gene dysregulation. Subsequently, we performed gene enrichment analysis using the DAVID tool to explore the biological processes, cellular components, molecular functions, and

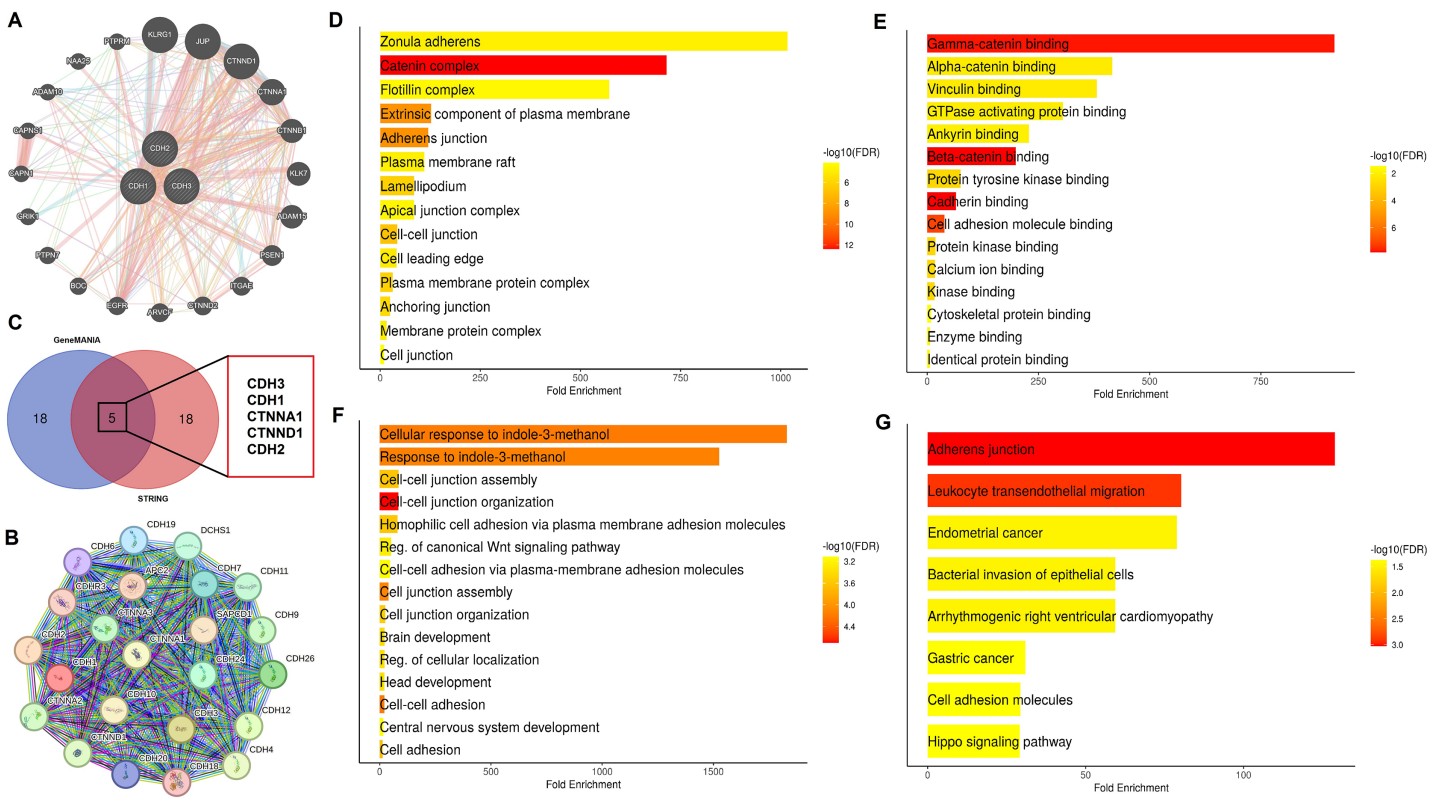

**Figure 7 Interaction networks and functional enrichment analysis of cadherin genes in NSCLC.** (A and B) Protein-protein interaction (PPI) networks of cadherin family genes (CDH1, CDH2, and CDH3) constructed using GeneMANIA STRING databases. (C) Venn diagram analysis identified five common genes (CDH1, CDH2, CDH3, CTNNA1, and CTNND1) present in both PPI networks. (D) Cellular component enrichment analysis (E) Molecular function enrichment analysis. (F) Biological process enrichment analysis. (G) Pathway enrichment analysis. *P*-value < 0.05.

pathways associated with these genes (Figs. 7D–7F). Gene enrichment analysis (Figs. 7D–7G) provided further insights into the biological significance of these cadherin genes. Cellular component enrichment (Fig. 7D) highlighted their association with structures such as the "zonula adherens, catenin complex, plasma membrane, and adherens junctions, emphasizing their role in maintaining cell-cell adhesion and integrity." Molecular function enrichment (Fig. 7E) revealed interactions with proteins involved in "gamma-catenin binding, beta-catenin binding, cadherin binding, and cytoskeletal protein binding, indicating their involvement in intracellular signaling and cytoskeletal organization." Biological process enrichment (Fig. 7F) identified key roles for these genes in "cell junction assembly, cell adhesion, and the regulation of the canonical Wnt signaling pathway," which is known to contribute to cancer progression and metastasis. Additionally, pathway enrichment analysis (Fig. 7G) demonstrated significant associations with the "adherens junction pathway, leukocyte transendothelial migration, the Hippo signaling pathway, and pathways related to gastric and endometrial cancer," further supporting the role of cadherin genes in tumorigenesis and metastasis. However, as a next step, future mechanistic studies are essential to investigate how these cadherins modulate the aforementioned pathways at the molecular and cellular levels.

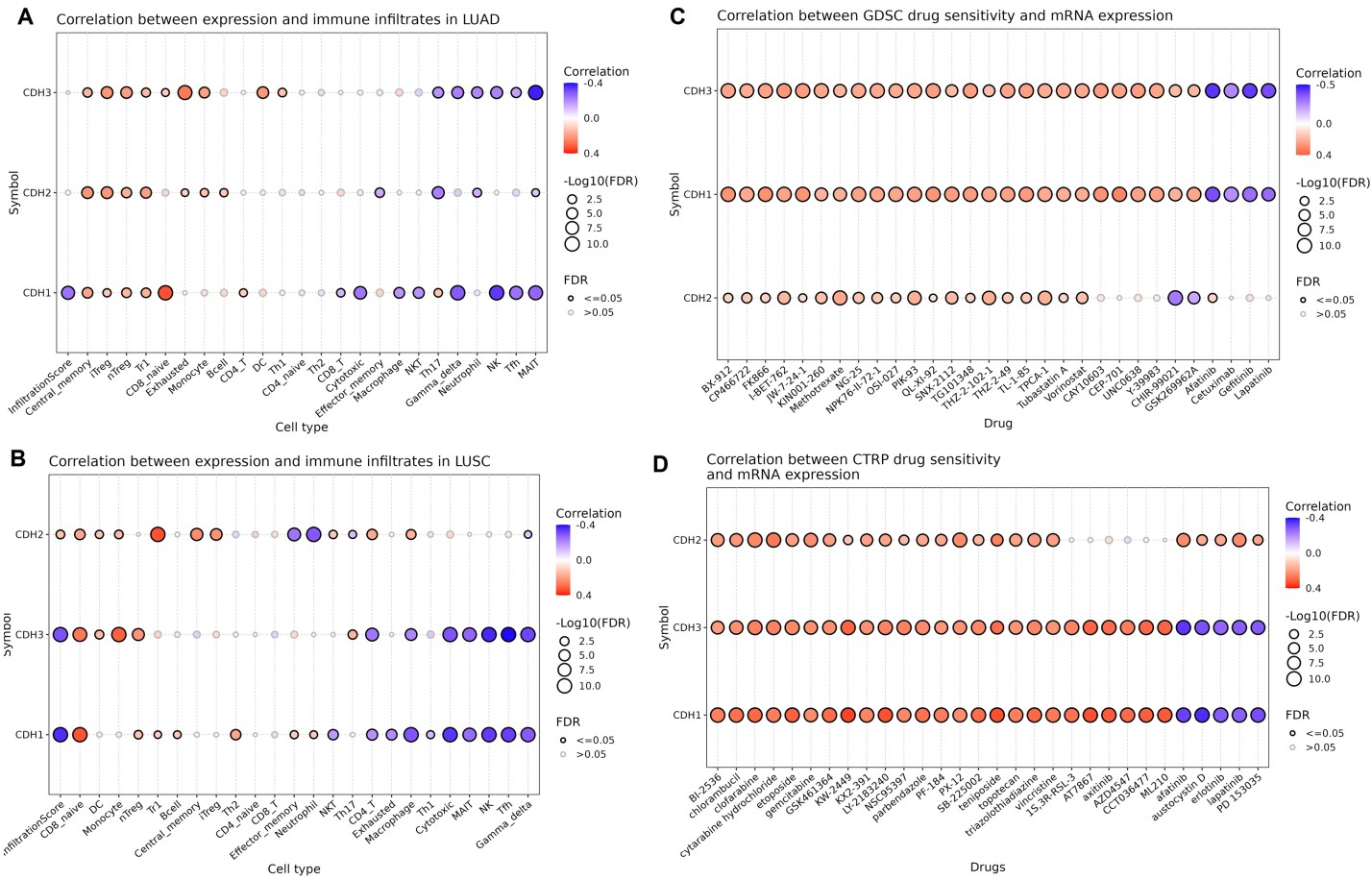

**Figure 8 Associations between cadherin gene expression, immune cell infiltration, and drug sensitivity in NSCLC.** (A and B) Correlation analysis of cadherin gene expression (CDH1, CDH2, and CDH3) with immune cell infiltration in LUAD and LUSC using the GSCA database. (C and D) Drug sensitivity analysis based on the GDSC and CTRP datasets. *P*-value < 0.05.

## Association of cadherin family gene expression with immune cell infiltration and drug sensitivity in NSCLC

Next, we investigated the relationship between the expression of cadherin family genes (CDH1, CDH2, and CDH3) and immune cell infiltration as well as drug sensitivity in NSCLC using the GSCA database. In LUAD (Fig. 8A), CDH3 showed a significant negative correlation with neutrophils and Th17 cells (Fig. 8A), while CDH2 exhibited a negative correlation with MAIT and neutrophils (Fig. 8A). CDH1, on the other hand, demonstrated significant negative correlations with CD8 and cytotoxic cells (Fig. 8A). In LUSC (Fig. 8B), CDH3 expression negatively correlated with cytotoxic, MAIT, and NK cells (Fig. 8B), while CDH2 displayed negative correlations effector memory and neutrophil cells (Fig. 8B). CDH1 showed relatively limited associations with immune cell types in LUSC.

In terms of drug sensitivity (Figs. 8C and 8D), CDH3 expression was significantly associated with resistance to multiple drugs. Across the GDSC dataset, CDH1, CDH2, and CDH3 expressions were correlated with resistance to BX-912, CP466722, FK866, and

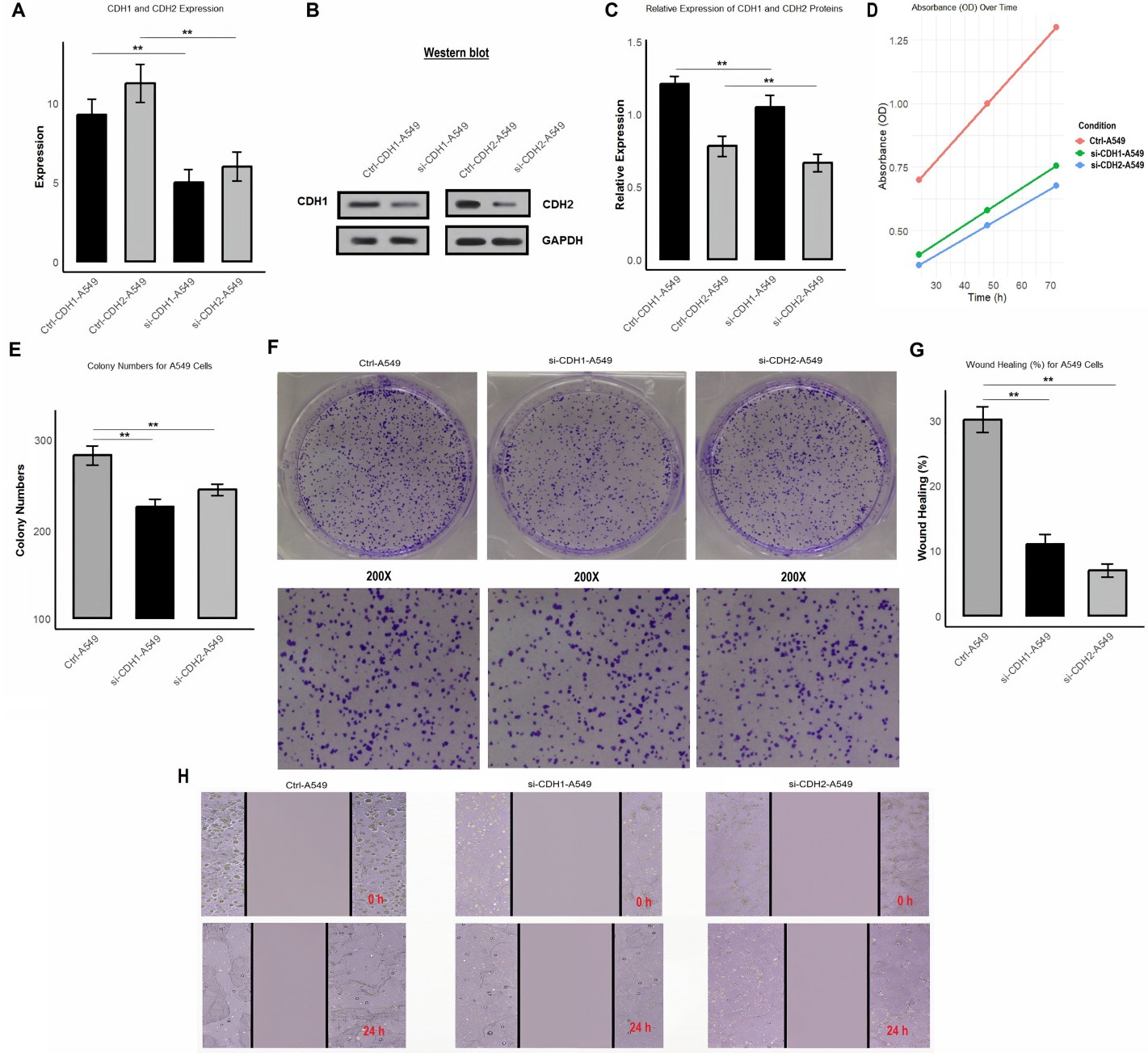

**Figure 9** **Silencing of CDH1 and CDH2 inhibits proliferation, colony formation, and migration in A549 cells.** (A) Validation of knockdown efficiency using RT-qPCR, showing significantly reduced mRNA levels of CDH1 and CDH2 in si-CDH1-A549 and si-CDH2-A549 cells, respectively, compared to control cells. (B and C) Western blot analysis confirming reduced protein expression of CDH1 and CDH2 after siRNA-mediated knockdown. (D) Proliferation assay results showing a significant reduction in OD values in si-CDH1-A549 and si-CDH2-A549 cells compared to controls. (E and F) Colony formation assay images (E) and quantified data (F) demonstrating a marked decrease in the number of colonies formed by si-CDH1-A549 and si-CDH2-A549 cells. (G) Representative wound healing images, (H) wound closure quantification for si-CDH1-A549 cells, and (I) wound closure quantification for si-CDH2-A549 cells. $P$**-value < 0.1.
I-BET-762 drugs (Fig. 8C) *etc*. Analysis using the CTRP dataset further validated these findings, showing consistent associations of CDH1, CDH2 and CDH3 expressions with resistance to drugs like chlorambucil, etoposide, and PF-184.

### Functional analysis of CDH1 and CDH2 knockdown reveals their roles in proliferation, colony formation, and migration in A549 cells

In the final part of the study, we investigated the functional role of CDH1 and CDH2 in A549 cells by performing gene knockdown using siRNA. We analyzed the impact of CDH1 and CDH2 silencing on their expression at both mRNA and protein levels, as well as on cell proliferation, colony formation, and wound healing. After knockdown, the efficiency was validated using RT-qPCR (Fig. 9A and Table S1) and Western blot analysis (Figs. 9B, 9C and Fig. S1). The mRNA and protein levels of CDH1 and CDH2 were significantly reduced in si-CDH1-A549 and si-CDH2-A549 cells, respectively, compared to control cells. Cell proliferation assay showed that knockdown of CDH1 and CDH2 led to a significant reduction in cell proliferation, as demonstrated by a decrease OD in transfected cells compared to control cells (Fig. 9D and Table S2). Colony formation assays further confirmed these findings, showing a substantial decrease in the number of colonies formed by si-CDH1-A549 and si-CDH2-A549 cells (Figs. 9E and 9F). Wound healing assays were conducted to assess the impact of CDH1 and CDH2 silencing on cell migration. Both si-CDH1 and si-CDH2 knockdowns resulted in significantly reduced wound closure percentages compared to control cells, as observed in the wound healing images and quantified data (Figs. 9G, 9H, and 9I). These results indicate that the silencing of CDH1 and CDH2 inhibits cell proliferation, colony formation, and migration in A549 cells, highlighting their potential roles in promoting tumorigenic and migratory properties in NSCLC.

## DISCUSSION

Non-small cell lung cancer (NSCLC), accounting for approximately 85% of all lung cancer cases, remains a leading cause of cancer-related mortality worldwide (*Leiter, Veluswamy & Wisnivesky, 2023*; *Hendriks et al., 2024*; *Xiang et al., 2025*). Despite advancements in diagnosis and treatment, the overall survival rate for NSCLC remains dismal due to its late detection, high heterogeneity, and propensity for metastasis (*Hendriks et al., 2024*; *Ouyang et al., 2025*). Cadherins, a family of transmembrane proteins primarily involved in cell-cell adhesion, have garnered attention for their roles in cancer progression, including EMT, a critical process in tumor invasion and metastasis (*Santarosa & Maestro, 2021*; *Fagotto & Aslemarz, 2020*; *Huang et al., 2024*). Among these, CDH1, CDH2, and CDH3 have been implicated in multiple cancers (*Van Roy, 2014*; *Ku et al., 2022*), yet their specific roles in NSCLC remain inadequately defined. Our study aimed to elucidate the expression, functional roles, and clinical significance of these cadherins in NSCLC, utilizing a comprehensive approach involving, *in silico*, and bioinformatics analyses.

Our findings demonstrate that CDH1, CDH2, and CDH3 were significantly upregulated in NSCLC cell lines and tissues compared to normal controls. This is consistent with prior

studies reporting overexpression of cadherins in various cancers, such as breast cancer (*Van Roy, 2014*) and colorectal cancer (*Abdelmaksoud-Dammak et al., 2017*), emphasizing their involvement in tumor progression. Notably, ROC curve analysis revealed that these genes exhibit exceptional diagnostic potential, with AUC values of 1.0, indicating their reliability as biomarkers for NSCLC. Similar diagnostic utility has been reported for CDH1 in breast cancer (*Xie et al., 2022*; *Huang, Ding & Yang, 2015*; *Corso et al., 2024*) and CDH2 in gastric cancer (*Zhao et al., 2021*; *Zhang et al., 2024*), supporting their potential as universal biomarkers across cancers.

In many cancer types, both E-cadherin (CDH1) and N-cadherin (CDH2) contribute to tumor behavior, though their functions can diverge depending on tumor type, microenvironment, and disease stage. For example, in prostate cancer, simultaneous modulation of E-cadherin and N-cadherin has been explored to inhibit EMT and metastatic progression (*Wang et al., 2017*; *Zhao et al., 2021*). Similarly, in gastric carcinoma, dual targeting of CDH1 and CDH2 suppressed migration and invasion (*Shenoy, 2019*). Another notable example is bladder cancer, where combined modulation of cadherin expression reduced invasiveness and improved drug sensitivity (*Lin et al., 2010*). Importantly, our data also emphasize a mechanistic complexity: while CDH1 is generally associated with epithelial integrity and EMT inhibition, CDH2 and CDH3 often promote mesenchymal features and EMT activation. This raises a significant therapeutic challenge—how to selectively target cadherins when they exhibit opposing roles. In contexts such as our findings (*e.g.*, Fig. 2F), where CDH1 suppresses and CDH2/3 promote EMT, strategies must aim to restore or stabilize CDH1 function while inhibiting CDH2/3-mediated signaling. Approaches such as isoform-specific inhibitors, RNA interference, or epigenetic modulation offer potential strategies, though concerns about off-target effects and disruption of normal tissue homeostasis must be addressed. Therefore, therapeutic strategies must be carefully designed to selectively inhibit pro-tumor cadherins while preserving or restoring the tumor-suppressive function of CDH1.

Epigenetic regulation of cadherins, particularly promoter methylation, has been previously noted as a critical mechanism in cancer biology (*Fan et al., 2019*; *Katto & Mahlknecht, 2011*; *Loh et al., 2019*; *Gracia, Sanchez-Laorden & Gomez-Sanchez, 2025*). Our findings of CDH1 and CDH3 promoter hypermethylation in NSCLC tumors, with relatively stable methylation for CDH2, are consistent with studies in gastric and prostate cancers (*Pistore et al., 2017*; *Berx & Van Roy, 2009*). The mutational landscape of cadherin genes in NSCLC, revealed by our study, also reflects previous findings of frequent CDH2 mutations in invasive cancers (*Guvakova et al., 2020*; *Tuersong et al., 2025*). However, our analysis identified a higher frequency of mutations in NSCLC compared to other cancers, suggesting that mutational burden may vary across cancer types. Furthermore, our study is among the first to identify specific SNP and CNV patterns in cadherins, providing novel insights into their genetic alterations in NSCLC. Furthermore, our study identified hsa-miR-217, hsa-miR-203a-3p.2, and hsa-miR-6766-3p as potential regulators of CDH1, CDH2, and CDH3 in NSCLC. RT-qPCR confirmed their significant downregulation in NSCLC cell lines, suggesting that reduced expression of

these miRNAs may contribute to cadherin dysregulation and promote tumor progression in NSCLC.

One of the most compelling aspects of our study is the correlation of cadherin expression with immune cell infiltration and drug sensitivity. CDH3's negative correlation with immune inhibitory molecules, such as CD160 and HAVCR2, highlights its potential role in modulating immune evasion in NSCLC. This finding aligns with reports in colorectal cancer, where cadherins influence the tumor microenvironment and immune response (*Harjunpää et al., 2019*; *Kasprzak, 2021*; *Xie et al., 2025*; *Tang et al., 2025*). Moreover, the association of cadherin genes with sensitivity and resistance to chemotherapeutic agents such as etoposide and chlorambucil offers new avenues for correcting the expression imbalance of these genes in NSCLC. While the concept of correcting the expression imbalance of cadherin genes presents an appealing therapeutic strategy, it is important to acknowledge that currently, no clinically approved treatments specifically target CDH1, CDH2, or CDH3 in NSCLC. Experimental approaches such as RNA interference (RNAi), antisense oligonucleotides, or CRISPR-Cas9-based gene modulation have shown potential in preclinical studies to selectively downregulate overexpressed genes; however, their translation into clinical therapies remains limited by delivery challenges and concerns over off-target effects. Therefore, future research should prioritize developing and validating cadherin-specific modulators with minimal toxicity. Additionally, any therapeutic strategy targeting cadherins must carefully consider their physiological roles in normal tissues, as broad inhibition could disrupt essential cell–cell adhesion processes and lead to unintended side effects. As such, the development of highly selective agents or context-specific delivery systems will be critical to minimizing adverse outcomes while leveraging the diagnostic and prognostic utility of cadherins in NSCLC. The observed negative correlations between cadherin gene expression and immune inhibitors, as well as immune cell infiltration, highlight a potential interplay between cadherins and the immune landscape in NSCLC. These findings imply that cadherins may influence tumor progression by affecting immune evasion mechanisms, which is critical in understanding the potential for immune checkpoint inhibitors as therapeutic options. However, it is important to stress that correlations observed in this study do not necessarily imply a causal relationship. While these correlations are promising, they do not establish a direct functional interaction between cadherin expression and immune modulation. It is possible that cadherins influence immune cell infiltration and immune checkpoint expression indirectly, potentially through their effects on tumor cell adhesion, EMT, or other aspects of tumor biology. To fully validate these findings and elucidate the mechanisms at play, further functional studies are needed. Experimental approaches such as immune cell co-culture systems, immune checkpoint inhibition assays, and *in vivo* tumor models could help establish a more direct link between cadherin expression and immune regulation.

Our results further demonstrate that CDH1 and CDH2 play significant roles in promoting tumorigenic processes in NSCLC. Silencing both cadherins in A549 cells led to reduced cell proliferation, colony formation, and migration, highlighting their

contribution to cancer growth and metastasis. The decreased proliferation observed after CDH1 and CDH2 knockdown aligns with their known roles in supporting cell survival and growth (*Alimperti & Andreadis, 2015*; *Nguyen & Martin, 2023*), with CDH2 in particular promoting migration through pathways like PI3K/AKT (*Alimperti & Andreadis, 2015*) and Rho GTPase (*Moghbeli, 2024*; *Maharati & Moghbeli, 2023*). The reduction in colony formation further supports the idea that cadherins are involved in anchorage-independent growth, a hallmark of cancer. In wound healing assays, both knockdowns resulted in impaired cell migration, reinforcing their roles in invasion and metastasis.

While our study provides significant insights into the roles of cadherin genes in NSCLC, it has certain limitations. The *in vitro* experiments were conducted in a limited number of cell lines, which may not fully represent the heterogeneity of NSCLC. Additionally, while our bioinformatics analyses provide robust correlations, functional validation of the identified pathways and immune interactions requires further experimental confirmation. Future studies should explore the therapeutic targeting of cadherin genes and their regulatory miRNAs, such as hsa-miR-217 and hsa-miR-6766-3p, to assess their potential in improving NSCLC outcomes.

## CONCLUSION

This study demonstrates that CDH1, CDH2, and CDH3 are significantly upregulated in NSCLC and are associated with tumor progression, poor prognosis, and advanced disease stages. They show strong diagnostic accuracy and are involved in key cancer-related pathways, including EMT and cell adhesion. Their expression correlates with immune cell infiltration, immune checkpoints, and drug resistance. Functional assays confirm their role in promoting proliferation and migration in NSCLC cells. Overall, cadherin genes serve as important biomarkers with potential relevance in NSCLC diagnosis and disease characterization.

### Funding

Funding was provided by the Henan Engineering Research Center for Key Immunological Biomaterials, the Key Scientific and Technological Project of Henan Province (242102310352) and the Scientific and Technological Research in Henan Province: Development of Rapid Diagnostic Reagent for Feline Infectious Peritonitis Virus Based on Microdroplet Single Cell Technology (252102110050). The funders had no role in study design, data collection and analysis, decision to publish, or preparation of the manuscript.

### Grant Disclosures

The following grant information was disclosed by the authors:
Henan Engineering Research Center for Key Immunological Biomaterials.
Key Scientific and Technological Project of Henan Province: 242102310352.
Scientific and Technological Research in Henan Province: 252102110050.

## Competing Interests

The authors declare that they have no competing interests.

## Author Contributions

- Quanzhong Yang conceived and designed the experiments, performed the experiments, prepared figures and/or tables, authored or reviewed drafts of the article, and approved the final draft.
- Nan Feng performed the experiments, prepared figures and/or tables, authored or reviewed drafts of the article, and approved the final draft.
- Feifei Shen analyzed the data, authored or reviewed drafts of the article, and approved the final draft.
- Lin Bai analyzed the data, authored or reviewed drafts of the article, and approved the final draft.
- Rihui Li analyzed the data, authored or reviewed drafts of the article, and approved the final draft.
- Shuang Li conceived and designed the experiments, authored or reviewed drafts of the article, and approved the final draft.
- Weikai Zhang conceived and designed the experiments, analyzed the data, authored or reviewed drafts of the article, and approved the final draft.

## Microarray Data Deposition

The following information was supplied regarding the deposition of microarray data:

Data is available at https://xenabrowser.net/datapages/ under gene names: CDH1, CDH2, and CDH3.

## Data Availability

- The original R code is available at: https://www.rdocumentation.org/packages/pROC/versions/1.18.5.
- The methylation data for this study is available at the TCGA database: https://xenabrowser.net/datapages/.
- The gene expression profile used in this research is available at NCBI: GSE199967.

## Supplemental Information

Supplemental information for this article can be found online at http://dx.doi.org/10.7717/peerj.19785#supplemental-information.

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
