# Peer review of "Prognostic role and functional impact of cadherin genes in non-small cell lung cancer tumorigenesis: mechanistic insights from in silico and in vitro analyses"

_PeerJ, doi:10.7717/peerj.19785_

## Round 0.1 · original submission · Major Revisions

Please address the concerns of all reviewers and revise the manuscript accordingly.

Reviewer 1 ·

Basic reporting

1. There appears to be an inconsistency in the description of significance thresholds in section 2.14 (e.g., P***-value <0.05 is repeated twice with different symbols). Are you sure P***-value <0.05?

2. Some survival analyses show trends without reaching statistical significance (e.g., CDH1 HR=1.3, p=0.083). Consider discussing the possible reasons for this and whether larger cohorts or meta-analyses could clarify this association.

3. The authors should consider discussion data on the molecular subtypes of each cell line (e.g., EGFR mutant, KRAS mutant, squamous vs adenocarcinoma) to better understand the representativeness and applicability of in vitro findings.

4. The manuscript nicely reports pathway enrichment and interactions, but direct mechanistic experiments (e.g., rescue experiments, signaling pathway analyses post-knockdown) are lacking. While possibly beyond the scope of this study, the authors should acknowledge this and outline plans for future mechanistic work.

5. The negative correlations between cadherin genes and immune inhibitory molecules and immune infiltrates are intriguing. However, correlation does not imply causation. Please elaborate more in the discussion on the potential implications and the need for functional immune assays to confirm these interactions.

6. Minor typographical and formatting issues are present (e.g., inconsistent gene symbol formatting, spacing). A thorough editorial review is necessary.

7. The description of the methodology in the manuscript is somewhat lacking in detail. It would be helpful if the authors could provide more specific information regarding the experimental procedures. Furthermore, citing previously published studies that have used similar experimental approaches (e.g., PMID: 38561760; PMID: 40098998) could enhance the reliability and validity of the methods presented, as it would provide additional support for the chosen techniques.

8. While the epidemiological background on lung cancer is informative, I would recommend updating some of the references in this section. The data presented may be a bit outdated. More recent studies would better reflect the current trends in lung cancer incidence and mortality. For example, the authors should consider citing the following recent publications: PMID38230766; PMID: 39726592. These sources provide up-to-date epidemiological data and could further enhance the contextual relevance of your study.

Experimental design

-

Validity of the findings

-

Reviewer 2 ·

Basic reporting

The provided raw data are not sufficient. Please see the detailed points in the additional comments.

Experimental design

-

Validity of the findings

Some of the conclusions seem overstated and are not fully supported by the presented data. Please see the detailed comments in the additional comments.

Additional comments

Yang et al performed bioinformatic analysis with the focus on cadherin genes in NSCLC. While the analyses appear comprehensive and wide-ranging, the rationale behind each analysis and the key insights they provide are not clearly articulated. Clarifying the purpose and interpretative value of these analyses would greatly enhance the manuscript's impact. The figures are neat and well-presented. However, there are incorrect labels and typos. Moreover, the data from in vitro experiments are not strong. Please see the detailed point below.

There are three main concerns.
The first concern is that the concept that cadherins can be potential therapeutic targets is not reasonable to this reviewer. The cadherins are important in normal biology, as the authors stated in the introduction. Therefore, targeting these genes/proteins will have off-target effects on normal cells.
The second concern is that the authors appear to be a little ambitious to draw a conclusion with three different cadherins. These cadherins are known to have different functions in different contexts. For example, during the EMT, the expression level of E-cadherin is reduced, whereas N-cadherin is increased. Therefore, it is unlikely that both E-cad and N-cad have a similar role in the same tumors. As the authors showed in Figure 2F, whereas N-cadherin and P-cadherin activated EMT, Ecad inhibited EMT. However, the authors only emphasized the activating role of N-cad/P-cad and did not mention the inhibitive role of E-cad. Although it is ok to emphasize the points based on the authors’ preference, it is potentially misleading to the readers who are not familiar with Cadherins.

The third concern is that some of the normal cell lines that were used in this study are not appropriate, and the conclusion may be damaged by inappropriate comparisons made between tumor cell lines and fibroblast cell lines. One would expect the comparison to be made between tumor cells and the normal counterparts (AT2 cells). However, in this study, two of the five normal cell lines are bronchial epithelial cells, whereas the other three cell lines are fibroblasts from lung tissues. MRC-5 (https://www.atcc.org/products/ccl-171), WI-38, (https://www.atcc.org/products/ccl-75), CCD19lu (https://www.atcc.org/products/ccl-210).

Other issues:
1. “This study aims to investigate the diagnostic, prognostic, and therapeutic potential of the Cadherin family genes.” This study did not really provide evidence to address whether cadherins have therapeutic potential. If the authors want to make such a claim, additional in vivo experiments are required. For example, implant the cadherin-expressing cells into mice and treat the mice with drugs that can target cadherins.
2. Figure. 1 B. It is highly recommended that the authors clarify what the inputs/datasets were used to generate the ROC curve, how and why the ROC curve was generated. It is unclear whether the ROC curve is an appropriate method to use here. To this reviewer, each plot appears to have two lines instead of a curve.
3. Figure 2C&D. The authors should clarify what input gene sets were used; if it is a customized gene set, the genes should be provided as a supplementary table. Adjusted NES, adjusted P-value, and FDR should be provided for each plot.
4. Figure 3 B&C. It is recommended the provide parameters used to select patients in the KM database. The number of patients is higher in KM than what is shown in Figure 3 B&C.
5. Figure 5A-B.
1) What happened to C5? Why was there no C5?
2) It is unclear why the authors looked into six molecular subtypes and what the take-home messages are from this analysis. The description and interpretation in the results section are confusing.
“The expression analysis revealed significant differences in cadherin gene (CDH1, CDH2, and CDH3) levels among the six molecular subtypes (C1 to C6) in both LUAD and LUSC (Figure 5A). This suggests that cadherin genes may play subtype-specific roles in NSCLC, contributing to differences in tumor biology and clinical behavior. ” It is highly recommended to clarify what the significant differences are and what the subtype-specific roles are.
3) The figure panels were cited incorrectly in section 3.5. E.g., “we examined its correlation with immune inhibitory genes across various cancers, as visualized in the heatmap analysis (Figure 5B). In LUAD and LUSC, differential correlations were observed between cadherin genes and key immune inhibitors. It should be Figure 5C.
6. Figure 6. The evidence is not strong enough to support the role of microRNA in regulating cadherins. It is recommended to overexpress microRNA in NSCLC cells and check the CHD1/2/3 expression.
7. The bioinformatic analysis in Figure 7 did not really provide any new information in the field. It is well known that cadherins physically interact with catenins, such as a-catenin, b-catenin, and delta-catenin. The authors should clarify the current state of knowledge regarding the interaction of cadherins and catenins in the field and discuss whether their findings are consistent with or divergent from previous studies
“To integrate these findings, we performed a Venn diagram analysis (Figure 7C) and identified five common genes (CDH1, CDH2, CDH3, CTNNA1, and CTNND1) that were consistently present in both PPI networks, suggesting their central role in NSCLC progression.”
The Venn diagram analysis basically showed the common genes between the two databases. It lacks a clear rationale for directly linking the identified common genes to “Their central role in NSCLC progression”.
8. The reviewer is concerned about Figure 9.
Figure 9A: What the authors provided is not real raw data.
Figure 9B & Supplementary Figure S1 do not make sense.
1) The molecular weight of CDH1 is around 135KD whereas that of CDH2 is around 140KD. However, the molecular weight of CDH1 is higher than CDH2 in the Figure.
2) Based on the labelling, Control-CHD2 reduced CDH1 expression. Si-CDH2 knocked out CDH1.
3) There is a typo: A54g should be A549.
4) Irrelevant sentences should not be shown. “I will send you….”
5) Figure 9C: What the authors provided is not real raw data. The three blots should be shown, and how the protein density was quantified should be provided.
6) Figure 9D: What the authors provide is not real raw data. Please provide the raw data from 24h, 48h, and 72h, as described in the method section. How absorbance was converted to the values in Figure 9D should be provided.
7) Figure 9E-F. The clones look weird. It is recommended to re-scan the plate/dish and show all the groups in the same image. Scale bars for each well should be provided.
8)Figure 9H. Why is the length of wounds so different at 0 hours?
si-CDH2-A549 bottom image, it seems that the wound was completely healed. The area between the two lines seems to be filled with cells.

Reviewer 3 ·

Basic reporting

Reviewer’s Comments:
This is a compelling study that explores the prognostic role and functional impact of cadherin genes in the tumorigenesis of non-small cell lung cancer, providing mechanistic insights through both in silico and in vitro analyses. However, there are several points that warrant further clarification or improvement.

Abstract

Line 15: The aim of the study is not stated in the abstract but is described in the final paragraph of the background section, where it is mentioned that the study aims to investigate the diagnostic, prognostic, and therapeutic potential of cadherin family genes (CDH1, CDH2, and CDH3) in non-small cell lung cancer (NSCLC). However, the study design (e.g., whether it is an experimental or observational study), the study setting or location, and the statistical analysis methods are not described in the methods section.

Background

Line 62: In the third paragraph of the background, the explanation of the research aim should be preceded by a brief narrative outlining the role of the cadherin family of genes in lung cancer.

Discussions

In the discussion section, it is recommended to include several studies that are relevant to and support the findings of this research.

Experimental design

Methods

1. Please provide a more comprehensive explanation of the research methods, including the study design, subject selection criteria, and study protocol.
2. The location of the study is not specified, and there is no mention of approval from a health research ethics committee.

Validity of the findings

-

---

## Round 0.2 · Minor Revisions

Please address the remaining concerns of Reviewer #2 and revise the manuscript accordingly.

Reviewer 1 ·

Basic reporting

The current manuscript is suitable for publication

Experimental design

good

Validity of the findings

good

Reviewer 2 ·

Basic reporting

Revised manuscript. No comment.

Experimental design

Revised manuscript. No comment.

Validity of the findings

Revised manuscript. No comment.

Additional comments

The manuscript has been improved through the revision. However, the responses to some questions/concerns are not satisfactory.

1) In response to the concern 1: the authors justified that “the goal is not to block their normal activity but to correct the imbalance caused by their overexpression in disease states, potentially reducing cancer growth and metastasis.” To this reviewer, this would be an ideal strategy for any overexpressed proteins in any type of cancers. However, the question is how to do so? If there are therapies/treatments to achieve this already, please be specific. Otherwise, please avoid overstating and discuss the potential off-target effect of such treatment in the manuscript.
2) In response to the concern 2: the authors justified that ” In many cancer types, both E-cadherin and N-cadherin can be simultaneously involved in modulating tumor behavior, though their exact contributions may vary depending on the tumor's microenvironment and stage.” Please add the point into discussion and provide a few examples where E-cad and noN-cad can be targeted simultaneously, references should be provided.
The authors should also discuss in a situation like Fig 2F, where CDH1 inhibits EMT and CDH2/3 activate EMT, how to selective target Cadherins.
3) Comment 3: Please clearly state that the controls are normal fibroblast cell lines in the manuscript.
4) Comments 4: If the authors agree that additional experiments are necessary but not resource to do it at the moment, please rephrase the sentence and the aims and do not claim the therapeutic potential.
5) Comment 5: The line 309-316 did not provide what values of NES, adjusted P value and FDR. The values should be added into the figure 2C-D. Eg. PMID: 35479111-Fig5)
6) Figure 5-A-B: C1,C2,C3,C4,C6 indicates there was subtype C5 (Also see line 166). If there were only five subtypes in this analysis, the authors need to provide explanation why C5 is not include here. In the figure legend, line 832, it was stated SIX MOLECULAR SUBTYPEs.
7) Comment 9: “The identification of common genes (CDH1, CDH2, CDH3, CTNNA1, and CTNND1) in both PPI networks supports their potential role in NSCLC progression,” The authors did not address the concern. The identification of common genes in both PPI networks only suggest these genes are associated with each other and in the same “network”. It does not support their potential role in NSCLC progression.
Suppl Figure_S1 still does not make sense. Pleas double check to see if the correct file was uploaded. Marker is required. It only takes one or two days to re-run WB. Loading markers are needed. Otherwise, how can one know the protein sizes are correct?

---

## Round 0.3 · accepted · Accept

All remaining issues were addressed and revised manuscript is acceptable now.